# Interplay of microRNAs and circRNAs in Epithelial Ovarian Cancer

**DOI:** 10.3390/ncrna10050051

**Published:** 2024-09-30

**Authors:** Heidi Schwarzenbach

**Affiliations:** Department of Gynecology, University Medical Center Hamburg-Eppendorf, 20246 Hamburg, Germany; hschwarzenbach@me.com; Tel.: +49-01622735795

**Keywords:** circRNAs, miRNAs, ceRNAs, ncRNAs, ovarian cancer, sponging, tumor progression

## Abstract

Epithelial ovarian cancer (EOC) with its high death incidence rate is generally detected at advanced stages. During its progression, EOC often develops peritoneal metastasis aggravating the outcomes of EOC patients. Studies on non-coding RNAs (ncRNAs), such as microRNAs (miRNAs) and circular RNAs (circRNAs), have analyzed the impact of miRNAs and circRNAs, along with their interaction among each other, on cancer cells. MiRNAs can act as oncogenes or tumor suppressors modulating post-transcriptional gene expression. There is accumulating evidence that circRNAs apply their stable, covalently closed, continuous circular structures to competitively inhibit miRNA function, and so act as competing endogenous RNAs (ceRNAs). This interplay between both ncRNAs participates in the malignity of a variety of cancer types, including EOC. In the current review, I describe the characteristics of miRNAs and circRNAs, and discuss their interplay with each other in the development, progression, and drug resistance of EOC. Sponging of miRNAs by circRNAs may be used as a biomarker and therapeutic target in EOC.

## 1. Introduction

Epithelial ovarian cancer (EOC) is the predominant ovarian cancer subtype and accounts for more than 95% of cases. The other non-epithelial subtypes include germ cell and sex-cord stromal cancer types, rare, small cell carcinoma, and ovarian sarcoma. EOC is histologically grouped into the following five major subtypes: high-grade serous, low-grade serous, clear cell, endometrioid, and mucinous ovarian cancer [1]. In 1973, the International Federation of Gynecology and Obstetrics (FIGO) staging system was introduced and revised in 1988 and 2013. There are four stages: FIGO I and II involve the restriction of the tumor to the ovaries and the tumor spread in the pelvis, respectively. FIGO III leads to the tumor spread outside the pelvis and its subtype FIGO IIIC defines the metastatic involvement of retroperitoneal lymph nodes, while FIGO IV covers distant metastases [2]. The high death incidence rate of EOC is due to asymptomatic features leading to the late diagnosis of the majority of patients. About three-fourths of FIGO III/IV patients have relatively poor survival prospects. Currently, the relative 5-year survival is 44%. In contrast, early-stage disease is treatable in nearly 90% of women, even in patients with more aggressive histologic subtypes. In this respect, the American Cancer Society estimates the number of cases with a new diagnosis of ovarian cancer of about 19,680 women in the United States for 2024, while 12,740 women will die from ovarian cancer (URL: https://www.cancer.org/cancer/types/ovarian-cancer.html, accessed on 27 September 2024).

EOC progression is accompanied with epithelial–mesenchymal transition (EMT) in which epithelial cells acquire fibroblast-like mesenchymal features and cellular plasticity resulting in the loss of their cell polarity, and adhesion to adjacent cells and the basement membrane. In turn, these phenotypically changed cells migrate through the extracellular matrix and settle down in other organs. As multicellular spheroids, they invade either locally to adjacent tissues or to pelvic and distal organs within the peritoneal fluid or ascites [3].

Therapy decisions are based on disease stage, histology, and biology, and comprise surgery and chemotherapy. Chemotherapy usually applies the combination of two different types of drugs, and includes platinum compounds, such as cisplatin or carboplatin, and taxanes, such as paclitaxel or docetaxel. Moreover, the administration of inhibitors against poly (ADP-ribose) polymerase molecules (PARP), which share in the DNA damage-repair process, have led to treatment success in some recurrent patients [4]. 

So far, numerous investigations have shown that non-coding RNAs (ncRNAs) participate in different stages of cancer, namely initiation, progression, and metastasis. In particular, abundant studies on microRNAs (miRNAs) have been published and demonstrated the impact of miRNAs in EOC along with its treatment [5]. MiRNAs can modulate post-transcriptionally the gene expression, especially by their inhibitory activity through binding to the 3′ untranslational region (UTR) [6], leading to the repression of translation. However, they can also activate the transcription through binding to promoter or enhancer regions [7]. Their dissemination is ensured by the apoptotic and necrotic cells that release them into the blood circulation or other body fluids (called liquid biopsy) [8], where they exist either cell-free, or associated with Argonaut (AGO) proteins [9]. MiRNAs may also be actively secreted in exosomes, small extracellular vesicles, into the blood [10] which transfer them from cell to cell, to modulate the characteristics of the recipient cells. Hence, these exosomal miRNAs play an important role in cell-to-cell communication, to induce tumor progression and spread [11,12]. Circular RNAs (circRNAs) are also single-stranded ncRNAs but are created as a circular conformation via non-canonical splicing or back-splicing events [13]. They form an important crosstalk network with miRNAs and mRNAs, regulating a variety of cellular events including cell division, signaling pathways, cell mobility, and metastasis [14]. The majority of circRNAs have been detected in the cytoplasm where they act as competing endogenous RNAs (ceRNAs), leading to the inhibition of miRNA expression. 

The current review article deals with the regulatory network of miRNAs and circRNAs and their interplay between each other. Their impact on EOC development, and progression and possible therapy approaches using these ncRNAs are discussed.

## 2. ncRNAs

Approximately 80% of the genome is transcribed to ncRNAs [15]. They constitute subgroups of miRNAs, circRNAs, lncRNAs (long non-coding RNAs), siRNAs (small interfering RNAs), and piRNAs (PIWI-interacting RNAs). These ncRNAs mainly act as silencers by inhibiting translation through binding to their mRNA targets. They play an essential role in various stages of cancer, namely initiation, progression, and metastasis [16]. In particular, their roles in various stages of EOC and their potential use in therapies have been reported [17]. Their ability to interact with each other allows the sequestered ncRNAs to no longer bind to their mRNA target, leading to an undisturbed translation of the mRNAs performed by the ribosomes. The ceRNA activity results in the formation of a large-scale RNA regulatory network across the transcriptome, and is involved in numerous processes, such as DNA methylation, apoptosis, DNA damage repair, EMT, and multidrug resistance (MDR). It is dependent on the concentration and character of the ncRNAs; therefore, it is difficult to predict which interactions between ncRNAs take place and what impact they have on the signaling network and the cellular behavior. Thus, such a regulatory network may imply the formation of an altered cellular protein expression profile that should also be considered in pathological conditions [18]. 

## 3. Biogenesis and Characteristics of miRNAs

MiRNAs are a family of evolutionary conserved, small ncRNA molecules which, as mentioned above, post-transcriptionally inhibit the mRNA translation by binding to the 3′ UTR [6]. Remarkably, miRNAs can also activate mRNA translation by binding different target sites, including the DNA promoter [19].

Similar to protein-coding mRNAs, the 21–23 nucleotide-long miRNAs are typically generated by splicing, capping, and polyadenylating [20]. In the nucleus, RNA polymerase II transcribes the long primary miRNAs (pri-miRNAs) that are in turn cleaved by the endonuclease Drosha which associates with the RNA-binding protein DiGeorge syndrome critical region 8 (DGCR8), to generate the 60–70-nucleotide stem-loop precursor miRNAs (pre-miRNAs) [21]. In a Ran-GTPase dependent fashion, the pre-miRNAs are then transferred from the nucleus to the cytoplasm by exportin 5. The hairpin-like structure is further processed by DICER1 in association with the RNA-binding protein transactivation-responsive RNA-binding protein (TRBP) to produce the mature duplex miRNAs. One of the strands of the mature miRNAs is loaded into the miRNA-induced silencing complex (miRISC), containing DICER1 and AGO proteins [22], whilst the other (star) strand is usually degraded. However, occasionally, both strands can be integrated into RISC. Here, the interaction of miRNAs with mRNAs takes place by sequence complementary binding, inhibiting the translation into protein or degrading the mRNA. In addition, miRNAs may also be produced by alternative routes on which mirtrons bypass the Drosha processing step, whereby the premiRNA is produced by a splicing reaction. The subsequent steps within RISC are similar (Figure 1) [23].

In cancer, the expression of miRNAs is deregulated, either up- or downregulated, leading to aberrant signaling pathways. These modulated levels are associated with tumorigenesis, tumor progression, metastasis, and drug resistance. In this respect, miRNAs can act as oncogenes or tumor suppressors [24]. So far, numerous miRNAs have been reported in ovarian cancer [25,26]. For example, miR-21 inhibits PTEN expression, regulates PI3K/AKT signaling, reduces cell apoptosis, and increases cell proliferation in EOC [27]. The expression of miR-214 and miR-150 is abnormally upregulated in EOC and these miRNAs can also inhibit PTEN, downregulate the PTEN protein, and initiate the protein kinase B (AKT) pathway [28]. MiR-135a, miR-200c, miR-216a, and miR-340 regulate EMT and thus, modulate the invasiveness of EOC cells [29]. Elevated levels of miR-130a have been shown to circulate in the serum of early recurrence patients before elevated levels of serum marker CA125 are detected. The deregulation of miR-130a, which regulates the multidrug resistance 1 (MDR1) and PTEN gene expression, correlates with the development of cisplatin resistance [30]. The miR-200 family is associated with EMT and tumor angiogenesis [25,26]. The levels of exosomal miR-200b and miR-200c can distinguish between malignant and benign ovarian tumors and their increased levels are mainly observed in advanced EOC [31]. The exosomal levels of miR-200b correlate with patient overall survival, and influence cell proliferation and apoptosis [32]. In contrast, miR-340 and miR-377 act as tumor suppressors. They decrease EOC cell migration and invasion, downregulate the Wnt/β-catenin pathway, and inhibit EMT [33,34].

## 4. Biogenesis and Characteristics of circRNA

More than 100,000 circRNAs have been identified in human transcriptomes. They form covalently closed loop structures with no exposed 3′ and 5′ ends. Their stable ring structure prevents exonuclease-mediated degradation, so that their half-life of circRNAs of more than 48 h is much longer than that of linear RNAs with 10 h [35].

CircRNAs are classified in three subtypes, exonic circRNAs (ecRNAs), exon-intron circRNAs (EIciRNAs), and circular intronic RNAs (ciRNAs) [13]. Four classical models describe the formation of circRNAs [36], as follows: (1) In the lariat-driven circularization model, the pre-mRNA is folded in such a way that nonadjacent exons are in close proximity to each other, initiating exon-skipping and back-splicing. Additional splicing by removing introns of the formed EIciRNAs leads to ecRNAs [37]. (2) In the intron-pairing-driven circularization model, the base pairing of long flanking complementary introns, such as Alu elements, induces back-splicing in which a downstream 5′ splice site is joined and ligated with an upstream 3′ splice site by a 3′-5′ phosphodiester bond at the junction site. (3) In the RNA-binding protein (RBP)-driven circularization model, RBPs bind to flanking introns on both sides of an exon and unite them for back-splicing. (4) In the ciRNA biogenesis, intron lariats are degraded by debranching enzymes, but a particular pre-mRNA structure, consisting of the GU- and C-rich elements in the 5′ and 3′-end, allows introns to escape debranching and become ciRNAs [37] (Figure 1).

CircRNAs are usually expressed at low levels. However, one gene can produce multiple circRNAs. Because of their circle formation, they are more resistant to exonucleases than linear RNAs. As miRNAs, they circulate in the bloodstream and can be transported by exosomes [38]. They either promote gene transcription by binding to DNA polymerase II or inhibit mRNA translation. However, the majority of circRNAs regulate the gene expression by sponging miRNAs, serving as ceRNAs. Thus, circRNAs inhibit miRNA activity by competing binding sites [14,39]. They can also directly interact with circRNA-binding proteins; hence, they can regulate the translocation of certain proteins [40].

In the following paragraphs, circRNAs are described as a sponge for miRNAs in ovarian cancer.

## 5. circRNAs Serve as Sponge for miRNAs

To date, many research articles on the sponging of miRNAs by circRNAs have been published (Table 1). The interactions of circRNAs with miRNAs regulate cellular functions and control the development of EOC through their involvement with different signaling pathways. The competition of circRNAs with miRNAs allows the activation or repression of the downstream components of the signaling pathways (Figure 2) [41,42,43,44].

Thus, circRNAs sponge miRNAs; therefore, they abrogate the binding of miRNAs to their targets in important cancer-associated signaling pathways leading to tumor progression.

In the following, circRNAs which sponge one or several miRNAs, as listed in Table 1, have been arbitrarily selected and some of them have been considered and described in more detail with a focus on circRNAs which have been reported to interact with several miRNAs.

Cisplatin, a cytostatic drug which inhibits DNA replication, is usually applied in the first-line treatment of EOC. In their study, Gao et al. [123] investigated the interactions among circ_000784, miR-532-5p, and NFIB by a dual-luciferase reporter assay and determined the effect of circ_0007841 on cisplatin in a xenograft mouse model. They showed that circ_0007841 conferred cisplatin resistance through the miR-532-5p/NFIB axis. In order to promote cisplatin resistance, circ_0007841 acted as a sponge for miR-532, leading to the upregulation of NFIB, a member of nuclear factor I (NFI) family which stimulated DNA replication [143]. Huang et al. [118] demonstrated that circ_0007841 also acted as a ceRNA for another miRNA, namely miR-151-3p, to stimulate the expression of the RNA-binding protein MEX3C, leading to cell proliferation, migration, and invasion of EOC cells.

In EOC, plexin B2 (PLXNB2) expression is usually upregulated, and the silencing of PLXNB2 inhibits cell proliferation and invasion. As demonstrated by Liang et al. [96], circ_0013958 acted as a sponge for miR-637 to regulate the expression of PLXNB2. In EOC tissues and cells, upregulated circ_0013958 downregulated miR-637 resulting in the activation of PLXNB2. The knockdown of circ_0013958 impeded EOC development through modulating the miR-637/PLXNB2 axis.

So far, circ_0025033 has been found to be a sponge for miR-370, miR-184, and miR-532. Ma et al. [119] reported that circ_0025033 was upregulated in EOC. Its knockdown blocked tumor growth in vivo. Circ_0025033 affected the solute carrier family member SLC1A5, a mitochondrial glutamine transporter for metabolic reprogramming in cancer cells [144], via sponging miR-370-3p. In this process, SLC1A5 abolished the anti-ovarian cancer part of miR-370-3p and reduced glutamine metabolism. Xenograft models were established by Hou et al. [57] to determine the role of circ_0025033 in vivo. The knockdown of circ_0025033 or the U6 snRNA-associated Sm-like protein LSM4 blocked the ability of colony formation, migration, invasion, and glycolysis metabolism in EOC cells. In this context, circ_0025033 promoted the progression of EOC cancer by activating the expression of LSM4 via targeting miR-184. The interaction between miR-532-3p and circ_0025033 or forkhead box protein M1 (FOXM1), a critical proliferation-associated transcription factor [145], was examined using a pull-down assay and a dual-luciferase reporter assay by Huang et al. [131]. They detected that circ_0025033 upregulated the expression of FOXM1 by sponging miR-532-3p. Knockdown of exosomes containing circ_0025033 and derived from paclitaxel-resistant cells impaired the resistance of this cytostatic drug in recipient EOC cells.

Circ_0061140 has been shown to sponge miR-136, miR-361, and miR-761. Zhu et al. [132] investigated the effects of circ_0061140 on tumor formation and paclitaxel sensitivity in vivo by a tumor formation assay. Knockdown of circ_0061140 inhibited cell proliferation, migration, and invasion, and promoted cell apoptosis and paclitaxel sensitivity via sponging miR-136. An in vivo model was also conducted by Zhang et al. [58] using a xenograft mouse. Circ_0061140 facilitated tumorigenesis in vivo through inhibiting miR-361 to upregulate the expression of RAB1A, a small GTPase known for its role in vesicular trafficking [146]. In addition, Ma et al. [59] found that circ_0061140 expression was upregulated in EOC tissues and cell lines. Knockdown of circ_0061140 significantly suppressed the proliferation, migration, invasion, and angiogenesis. The oncogenic behavior of circ_0061140 referred to its ability to upregulate the expression of a mitochondrial inner membrane protein LETM1 by sponging miR-761.

Circ_0078607 has been shown to be a ceRNA for miR-32, miR-196, and miR-518a and to usually act as a tumor suppressor gene. In their study, Jin et al. [56] detected that circ_0078607 and salt inducible kinase 1 (SIK1) were downregulated in EOC tissues and cells. Overexpressed circ_0078607 suppressed EOC cell proliferation, migration, invasion, and promoted apoptosis by sponging miR-32-5p leading to the upregulation of SIK1, a target of miR-32-5p. The inhibitory effect of circ_0078607 on EOC progression could be reversed by the silencing of SIK1, a protein kinase which targets major plasma membrane transporters, such as the Na(+)/K(+)-ATPase and Na(+)/H(+) exchangers [147]. Furthermore, Dai et al. [122] showed that the overexpression of circ_0078607 inhibited cisplatin resistance in nude mice by sequestering miR-196b-5p to upregulate GAS7, while Zhang et al. [60] confirmed that circ_0078607 was downregulated in EOC. Bioinformatics and luciferase reporter analysis identified miR-518a-5p as a target of circ_0078607, while Fas is a target of miR-518a-5p. Thus, circ_0078607 suppressed EOC progression by sponging oncogenic miR-518a-5p to induce expression of the cell death molecule Fas which triggered apoptosis [148].

Circ_CELSR1 has been shown to interact with miR-149, miR-598, and miR-1252. Using paclitaxel-resistant EOC cells and tissues, Wei et al. [134] revealed that, the levels of circ_CELSR1 were upregulated, and its knockdown increased paclitaxel sensitivity and cell apoptosis, whereas inhibiting cell viability, colony formation and cell cycle process of resistant EOC cells. In a murine xenograft model assay, circ_CELSR1 silencing impeded paclitaxel resistance by regulating the miR-149-5p/SIK2 axis. Accordingly, circ_CELSR1 positively modulated SIK2 expression via sponging miR-149-5p. The increased SIK2 expression resulted in EOC progression, and might permit a treatment approach through regulating cellular metabolism, comprising glucose and lipid metabolism [149]. In addition, Zhang et al. [135] found that the paclitaxel resistance of EOC by circ_CELSR1 can also occur by regulating the expression of the protooncogene FOXR2 via miR-1252. Furthermore, Zeng et al. [107] used an abdominal cavity metastasis nude mouse model to analyze the in vivo function of circ_CELSR1. They showed that knockdown of circ_CELSR1 suppressed proliferation, migration, invasion, and EMT, but stimulated apoptosis in EOC cells, and also suppressed EOC growth and metastasis in nude mice. These effects were reversed by the inhibition of miR-598 or overexpression of BRD4. BRD4 is a member of the bromodomain and extra-terminal (BET) protein family and plays a role in super-enhancer organization and oncogene expression regulation [40].

Finally, in EOC tissues, circ_MYLK levels are significantly higher than those in adjacent tissues and its expression is remarkably associated with pathological staging and poor prognosis in EOC patients. Accordingly, circ_MYLK may promote the malignant progression of EOC through the downregulation of miR-652 [86].

## 6. Sponging of a miRNA by Several circRNAs

As described above, circRNAs can sponge several miRNAs, whereas conversely, several circRNAs can interact with one miRNA. In the following, single miRNAs which are sponged by several circRNAs, as listed in Table 2, are arbitrarily selected and some of them considered and described in more detail.

MiR-145 has been exposed to be a target for circ_0009910, circ_0015756, circ_ITCH, circ_VPS13C, and circ_WHSC1. In their study, Li et al. [52] confirmed that miR-145 was bound by circ_0009910 which negatively regulated miR-145. MiR-145 reversed the biological function of circ_0009910 in proliferative and motile phenotypes, and active status of the proinflammatory signaling pathway of NF-κB [150] and the pathway of Notch which is involved in angiogenesis, stem cell maintenance, and EMT [151]. Using nude mice, Pan et al. [53] showed that circ_0015756 which was highly expressed in EOC cells and promoted the tumor growth via the miR-145-5p/PSAT1 axis. In mechanical analysis, circ_0015756 directly bound to miR-145-5p which targeted the phosphoserine aminotransferase PSAT1, accelerating EOC tumorigenesis. Moreover, Lu et al. [101] demonstrated that the treatment of EOC cells with the anesthetic substance propofol suppressed the cell viability, cycle, and motility whereas elevating the apoptosis rate, and upregulated miR-145 in a dose-dependent manner. The anti-tumor role of propofol is partly owed to the upregulation of miR-145 which is a direct target of circ_VPS13C. Thus, propofol suppressed the progression of EOC cancer through the upregulation of miR-145 via suppressing circ_VPS13C. Zong et al. [82] showed that circ_WHSC1 was upregulated in EOC tissues, and increased cell proliferation, migration, and invasion, whereas it inhibited cell apoptosis. It sponged miR-145, leading to the upregulation of the expression of downstream targets, namely the transmembrane glycoprotein mucin 1 and the telomerase reverse transcriptase hTERT. Exosomes transferred circ_WHSC1 to peritoneal mesothelial cells, promoting peritoneal dissemination. Conversely, Hu et al. [71] reported the suppressive role of circ_ITCH function in the malignant progression of EOC in vitro and in vivo. Circ_ITCH acted as a ceRNA to sponge miR-145, increasing the level of the Ras p21 protein activator RASA1, a regulator of Ras GDP and GTP, which are involved in angiogenesis, cell proliferation, and apoptosis [152].

MiR-370 serves as a sponge for circ_0025033, circ_0070203, and circ_0000714. Ma et al. [119] reported that circ_0025033 affected the mitochondrial glutamine transporter for metabolic reprogramming SLC1A5 in EOC cells [144], via sponging miR-370-3p, resulting in the abolition of its anti-ovarian cancer role. In serous ovarian cystadenocarcinoma (HGSOC) FIGO stages III-IV, Tang et al. showed that circ_0070203 could upregulate the expression of the TGFβ receptor 2 via sponging miR-370-3p using cell lines and tissues. Overexpression enhanced the migrative, invasive abilities of EOC cells though the expression of EMT-related proteins [3]. Using a microarray, Guo et al. [130] detected that RAB GTPase family proteins were significantly overexpressed in paclitaxel-resistant EOC cells. Circ_0000714 acted as a sponge for miR-370-3p, and increased the expression of the Ras-related protein RAB17 through the CDK6/RB signaling pathway, which plays a role in the malignant progression of paclitaxel-resistant EOC cells.

MiR-532 has been shown to be inhibited by circ_0007841, circ_0025033, and circ_ATP2B4. In a xenograft mouse model, Gao et al. [123] showed that circ_0007841 conferred cisplatin resistance through the miR-532-5p/NFIB axis. In an interrelated manner, Huang et al. [131] detected that circ_0025033 upregulated the expression of the proliferation-associated transcription factor FOXM1 by sponging miR-532-3p, playing a role in paclitaxel resistance. Wang et al. [109] reported the positive correlation of upregulated circ_ATP2B4 with EOC progression. Exosomes transmitted circ_ATP2B4, which acted as the ceRNA of miR-532-3p, to infiltrated macrophages, to relieve the repressive effect of miR-532-3p on its target of the sterol regulatory element-binding factor SREBF1. In addition, circ_ATP2B4 induced macrophage M2 polarization by regulating the miR-532-3p/SREBF1/PI3Kα/AKT axis, thereby resulting in immunosuppression and EOC metastasis in vitro and in vivo.

Finally, circ_0010467, circ_0013958, and circ_0051240 have been revealed to be ceRNAs for miR-637. Wu et al. [121] observed an increased expression of circ_0010467 in platinum-resistant EOC cells, tissues, and serum exosomes, and a positive correlation with advanced tumor stages and the poor prognosis of EOC patients. The AU-rich element RNA-binding protein AUF1 stimulated circ_0010467 to promote platinum resistance through inducing tumor cell stemness, while circ_0010467 acted as a miR-637 sponge to activate the leukemia inhibitory factor (LIF) in the miR-637/LIF/STAT3 axis. Liang et al. [96] found that upregulated circ_0013958 downregulated miR-637 resulting in the activation of the surface receptor PLXNB2 in EOC cells. Knockdown of circ_0013958 impeded EOC development through modulating the miR-637/PLXNB2 axis. Zhang et al. [103] observed significantly increased levels of circ_0051240 in EOC tissues. Circ_0051240 acted as a sponge for miR-637 which targeted the kallikrein-related peptidase 4 KLK4 mRNAs in EOC cells. It promoted EOC cell proliferation, migration, and invasion in vitro, while it stimulated tumor formation in vivo.

## 7. EOC Therapies

The current clinical standard treatment strategy usually includes tumor cytoreductive surgery followed by platinum and paclitaxel chemotherapy. Platinum, such as cisplatin and carboplatin, covalently binds to purine bases and so introduces DNA damage, including monoadducts or inter- and intra-strand crosslinks. This leads to the interference of the replication machinery, G2/M cell arrest, and cell death by apoptosis or necrosis [153]. Paclitaxel additionally disrupts cell division by impeding the breakdown of the spindle apparatus during mitosis. As a result, the mitosis remains incomplete with no reproduction of the cells by the non-distributed DNA [154].

However, frequently, patients succumb to chemotherapeutic resistance and recurrence, even within several years after the initial treatment with these drugs. Chemotherapy resistance is one of the main obstacles in cancer treatment. Despite the improved chemotherapy regimens, such as intraperitoneal delivery and target therapies including poly (ADP-ribose) polymerase (PARP) inhibitors and antiangiogenic agents, these methods have only somewhat contributed to extend the 5-year survival rate in advanced EOC. To address the problem of drug resistance and extend the 5-year survival rate, PARP inhibitors have been coupled with ferroptosis, a type of programmed cell death dependent on iron which accumulates lipid peroxides [155]. Likewise, the response rates to immunotherapy with immune checkpoint inhibitors (ICIs), chimeric antigen receptor (CAR)-, and T cell receptor (TCR)-engineered T cells among EOC patients remain modest. Newly developed therapeutic targets that utilize nanomedicine technology provide new chances for the treatment of EOC patients, and might have the potential to prolong patient survival. However, the efficacy of such drugs may be accompanied with hyper-progressive disease and the toxicity of the treatments [156].

It has been reported that changes in the canonical Wnt/β-catenin and the Notch signaling pathways are relevant in EOC development, progression, and resistance. In the Wnt/β-catenin signaling pathway, Wnt proteins bind to receptors of the Frizzled and low-density lipoprotein receptor-related protein families on the cell surface. The generated signal is transferred through several cytoplasmic components until β-catenin, which in turn enters the nucleus to form a complex with the transcription factor TCF and activates the transcription of the Wnt target genes. Modulations of the Wnt/β-catenin signaling pathway include mutations in β-catenin or other key pathway members, as well as hypermethylation and silencing of gatekeeper antagonists, or overexpression of Wnt ligands or receptors. They lead to increased cancer cell proliferation and migration [157]. The Notch signaling pathway is activated by a receptor-ligand binding between two neighboring cells, leading to a conformational change in the Notch receptor. Following two cleavages, the Notch intracellular domain is transferred into the nucleus, where it binds to ubiquitous transcription factor CSL and converts a large co-repressor complex into a transcription activating complex. As a result, the transcription of Notch target genes is stimulated, among others p21, cyclin D1 and 3, c-myc, and members of the NF-κB family, which regulate proliferation, differentiation and apoptosis [151].

In particular, the modulation of these signaling pathways has led to the development of therapeutics that target these pathways. Accordingly, the development of new therapies using ncRNAs, such as miRNAs or circRNAs, may be of strategical interest.

## 8. circRNAs as Predictive Biomarkers

Due to the lack of an effective early detection screening test, the majority of EOC patients are initially diagnosed with advanced disease. CircRNAs play important roles in cancer tumorigenesis and progression, and represent prognostic biomarkers. Therefore, they could be eligible for a screening test. To date, there are numerous studies on circRNAs that describe circRNAs as biomarkers in tissues, plasma, serum and exosomes for EOC (Table 3).

## 9. circRNAs as Therapeutic Agents and Targets

So far, most studies have focused on the development of vaccines and nucleic acid-derived drugs considering miRNAs as therapeutic markers. In this respect, numerous investigations have taken advantage of the inhibitory effects of miRNAs on translation. MiRNAs can have oncogenic or/and tumor suppressive behavior. Restoring the miRNA tumor-suppressive function and inhibiting the oncogenic function have been considered to improve cancer treatment. Hence, assays, such as synthetic tumor suppressive miRNA mimics and anti-oncogenic agomiRs that bind miRNAs to agonist their oncogenic potential, have been developed [172,173,174]. However, circRNAs mainly act as a sponge for miRNAs, and so may serve as agomiRs, disturbing the function of miRNAs. Therefore, the interplay between miRNAs and circRNAs should be considered.

To date, the application of circRNAs in potential therapy approaches has been reviewed by Holdt et al. [175]. The chemistry, manufacturing, and controls (CMCs) process for circRNAs is comprised of the following four steps: (1) plasmid construction and proof of concept; (2) in vitro synthesis and purification; (3) circularization (spliceosomes and ligases) and purification; and (4) encapsulation and partitioning. Different methods to either overexpress or knockdown circRNAs in vitro and in vivo have been established [176]. Overexpression of circRNAs can be achieved by providing plasmids containing circRNA-producing exons. They can also be synthesized as miRNA mimics by applying RNA ligases or ribozymes and introducing photolabile linkers. In addition, synthetic circRNAs can be engineered to contain several miRNA-binding sites. They are transcribed and cyclized in vitro and can be delivered to specific tissue with adeno-associated virus (AAV) [177,178]. Chemical modifications of circRNAs improve stability and binding affinity, while coatings of circRNAs with proteins facilitates their recognition by cancer cells. In contrast, silencing of circRNAs can be achieved by siRNA, short hairpin RNAs (shRNA), CRISPR/Cas9, and RNA-targeting Cas13 system [179,180]. Typically, shRNAs and siRNAs that are 20–25 base pairs, double stranded RNA molecules operate via the RNA interference, and have been widely applied in functional studies to inhibit oncogenic circRNAs in vitro and in vivo [179]. It is worth mentioning that the CRISPR/Cas9 system established the first circRNA knockout mice model to knockdown circRNAs without apparent off-target effects. In addition, the CRISPR/Cas13 system successfully achieved the knockdown of circRNA expression [180].

In order to efficiently deliver circRNAs or their inhibitors to the target organs, various techniques have been employed including nanoparticles, adenoviruses, and plasmids [17,181,182,183,184]. Nanoparticles which are frequently used can be produced by organic materials, such as liposomes, polymers, and dendrimers or inorganic materials, such as gold and metal oxides [185].

## 10. Conclusions

Aberrant signal transduction pathways lead to tumor progression and metastasis and are excellent targets for therapeutic approaches that permit the inhibition of oncogenic signals. Therefore, the eligibility of ncRNAs that modulate these pathways should be considered for the use of therapeutic agents as well as biomarkers since they may reflect the dynamic of the disease course, including recurrence and resistance to therapy. The interplay between circRNAs and miRNAs provides new information that may contribute to understand the complex biological network of ncRNAs in the regulation of the mechanisms underlying EOC pathogenesis. The establishment of biological network models in which ncRNAs, e.g., circRNAs and miRNAs as well as lncRNAs, are involved have to be created to design EOC-associated signatures that allow their application for screening or treatment decisions by the physician. However, the multiple interactions between circRNAs and miRNAs should be taken into account. One circRNA can sponge several miRNAs and several circRNAs can sponge one miRNA. In addition, miRNAs can also be inhibited by other ncRNAs, such as lncRNAs. In addition, one signaling pathway can be disturbed by several ncRNAs. Conversely, one ncRNA can affect multiple signal pathways that may even cross-talk among each other. This complex interplay among ncRNAs and signaling pathways makes it particularly difficult to exactly define the effectiveness of a treatment with these molecules in a larger context. Therefore, before developing a targeted therapy using circRNAs to suppress miRNAs, their manifold behavior and interaction in both time and cellular location should be analyzed. Consequently, this should also include a circRNA as a therapeutic agent, which is considered to inhibit an oncogenic miRNA, and may also potentially inhibit tumor suppressive miRNAs.

Furthermore, researchers have identified numerous targets of miRNAs across different pathways within the same tumor types, including EOC. One miRNA is able to bind to a broad range of mRNA targets, leading to the inhibition and modulation of the expression of a variety of proteins, involved in both oncogenic and tumor suppressive functions and in different signaling pathways. These off-target effects mediated by miRNAs may result in potential non-specific toxicity and elicit unintended cellular and harmful immune responses that may accompany the treatment outcome. MiR-34 is an example for restoring tumor-suppressive functions along with non-specific toxicity. This miRNA is significantly downregulated in different cancer types including EOC and inhibits multiple oncogenic pathways [186]. A first-in-human phase I clinical trial (NCT01829971) was conducted to investigate the safety, pharmacokinetics, and clinical activity of a liposomal formulation of an miR-34 mimic (known as MRX34). The clinical study was stopped by the sponsoring company (Mirna Therapeutic, Inc.) since the patients dosed with MRX34 displayed multiple immune-related severe adverse effects after initial good tolerability [187]. Thus, designing tumor suppressive miRNA mimics or inhibitors for oncogenic miRNAs with a better specificity can help mitigate off-target effects. In addition, the technical platforms and the selection of an appropriate delivery system have to be advanced. New possibilities provide exosomes manipulated with specific surface marker to direct then to the cancer cells. To address these issues and assure that patients do not suffer from substantial adverse side effects, the development of an assay using a circRNA as a sponge for miRNAs may be a laborious endeavor.

In summary, the decoding of the regulatory network of ncRNAs remains an important task for the future to develop effective therapeutic agents to inhibit tumor progression and to overcome drug resistance.

## Figures and Tables

**Figure 1 ncrna-10-00051-f001:**
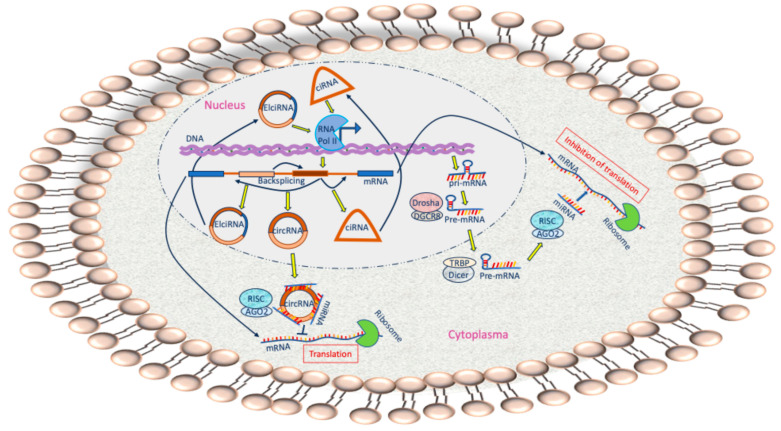
Biogenesis and function of miRNAs and circRNAs. Important components of the biogenesis of miRNAs and circRNAs, as described in more detail in the text, are depicted. As shown, EIciRNAs and ciRNAs promote the transcription by RNA polymerase II (RNA Pol II). MiRNAs bind to and inhibits the translation of mRNA, while the binding of circRNAs to miRNAs abrogates the inhibition, resulting in the translation of mRNAs by the ribosome.

**Figure 2 ncrna-10-00051-f002:**
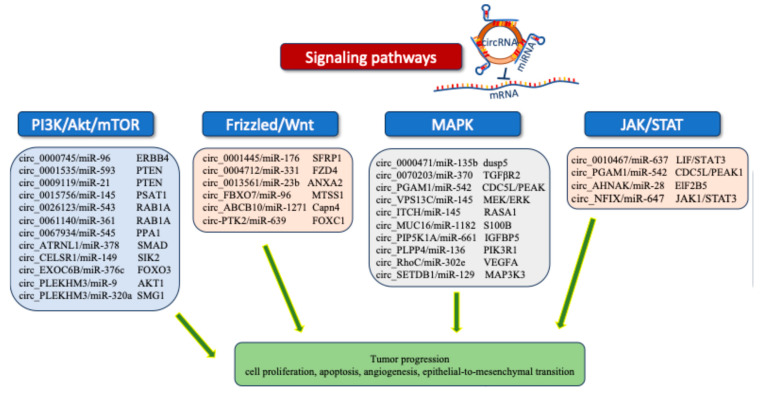
CircRNA/miRNA interactions and their involvement in different signaling pathways.

**Table 1 ncrna-10-00051-t001:** Interactions of circRNAs and miRNAs and their targets in EOC, arranged according to their function.

circRNAs	miRNAs	Targets	Function		Ref.
			* Increasing	** Decreasing	
circ_0000144	miR-610	ELK3	progression	-	[45]
circ_0000471	miR-135b	dusp5		progression	[46]
circ_0001445	miR-576	SFRP1	-	progression	[47]
circ_0004712	miR-331	FZD4	progression	-	[48]
circ_0007444	miR-23a	Dicer1	-	progression	[49]
circ_0007615	miR-874	TUBB3	progression	-	[50]
circ_0007874	miR-760	SOCS3	-	progression	[51]
circ_0009910	miR-145	-	progression	-	[52]
circ_0015756	miR-145	PSAT1	progression	-	[53]
circ_0021573	miR-936	CUL4B	progression	-	[54]
circ_0072995	miR-122	SLC1A5	progression	-	[55]
circ_0078607	miR-32	SIK1	-	progression	[56]
circ_0025033	miR-184	LSM4	progression	-	[57]
circ_0061140	miR-361	RAB1A	progression	-	[58]
	miR-761	LETM1	progression	-	[59]
circ_0070203	miR-518a	Fas	-	progression	[60]
circ_ATRNL1	miR-152	-	progression	-	[61]
circ_BNC2	miR-223	LARP4	-	progression	[62]
	miR-223	FBXW7	-	progression	[63]
circ_AHNAK	miR-28	EIF2B5	-	progression	[64]
circ_CDR1	miR-135b	-	-	progression	[65]
circ_CRIM1	miR-383	ZEB2	progression	-	[66]
circ_EPSTI1	miR-942	-	progression	-	[67]
circ_EXOC6B	miR-421	RUS1	-	progression	[68]
circ_FAM53B	miR-646	VAMP2	progression	-	[69]
	miR-647	MDM2	progression	-	[69]
circ_FBXO7	miR-96-5p	MTSS1/Wnt	progression	-	[70]
circ_ITCH	miR-145	RASA1	-	progression	[71]
circ_MFN2	miR-198	CUL4B	progression, glycolysis	-	[72]
circ_MUC16	miR-1182	S100B	progression	-	[73]
circ_NFIX	miR-647	JAK1/STAT3	progression	-	[74]
circ_PGAM1	miR-542	CDC5L/PEAK1	progression	-	[75]
circ_PHC3	miR-497	SOX9	progression	-	[76]
circ_PIP5K1A	miR-661	IGFBP5	progression	-	[77]
circ-PTK2	miR-639	FOXC1	progression	-	[78]
circ_RNF144B	miR-342	FBXL11	progression	-	[79]
circ_RhoC	miR-302e	VEGFA	progression	-	[80]
circ_SETDB1	miR-129	MAP3K3	progression	-	[81]
circ_WHSC1	miR-145	MUC1	progression	-	[82]
circ_WHSC1	miR-1182	hTERT	progression	-	[82]
circ_ZNF608	miR-152	-	progression	-	[61]
circ_UBAP2	miR-144	-	-	progression	[83]
circ_PLEKHM3	miR-9	BRCA1/DNAJB6/KLF4/AKT1	-	progression	[84]
	miR-320a	SMG1	-	progression	[85]
circ_MYLK	miR-652	-	progression	-	[86]
circ_0001741	miR-188	FOXN2	-	proliferation	[87]
circ_0004390	miR-198	-	proliferation	-	[88]
circ_ITCH	miR-10a	-	-	proliferation	[89]
circ_PVT1	miR-149	-	proliferation	-	[90]
circ_UBAP2	miR-382	PRPF8	proliferation	-	[91]
circ_ASXL1	miR-320d	RACGAP1	proliferation, migration	-	[92]
circ_CDK17	miR-22	CD147	proliferation, migration	-	[93]
circ_0001535	miR-593	PTEN	-	proliferation, migration [94]	
circ_0000554	miR-567	-	proliferation, invasion	-	[95]
circ_0013958	miR-637	PLXNB2	proliferation, invasion	-	[96]
circ_ABCB10	miR-1271	Capn4/Wnt	proliferation, invasion	-	[97]
circ_MTO1	miR-182	KLF15	-	proliferation, invasion	[98]
circ_RHOBTB3	miR-23a	-	-	proliferation, invasion	[99]
circ_9119	miR-21	PTEN/Akt	-	cell viability	[100]
circ_VPS13C	miR-145	MEK/ERK	cell cycle, motility	-	[101]
circ_PLOD2	miR-378	-	propagation	-	[102]
circ_0051240	miR-637	KLK4	migration, invasion	-	[103]
circ_CSPP1	miR-1236	-	invasion, migration	-	[104]
circ_NFIX	miR-518a	TRIM44	angiogenesis	-	[105]
circ_0026123	miR-124	EZH2	proliferation, metastasis	-	[106]
circ_CELSR1	miR-598	BRD4	proliferation, metastasis	-	[107]
circ_ATRNL1	miR-378	Smad4	-	angiogenesis, metastasis	[108]
circ_ATP2B4	miR-532	SREBF1	metastasis	-	[109]
circ_IFNGR2	miR-378	ST5	metastasis	-	[110]
circ_0002711	miR-1244	ROCK1	cell growth, glycolysis	-	[111]
circ_0005585	miR-23a/b/ 15a/15b/16	ESRP1	colonization	-	[112]
circ_0070203	miR-370	TGFβR2	EMT	-	[113]
circ_0013561	miR-23b	ANXA2	EMT	-	[114]
circ_FGFR3	miR-29a	E2F1	EMT	-	[115]
circ_S-7	miR-641	ZEB1, MDM2	EMT	-	[116]
circ_0000745	miR-3187	ERBB4/PI3K/AKT	cell stemness	-	[117]
circ_0007841	miR-151	MEX3C	development	-	[118]
circ_002503	miR-370	SLC1A5	development	-	[119]
circ_CERS6	miR-630	RASSF8	-	development	[120]
circ_0010467	miR-637	LIF/STAT3	platinum resistance	-	[121]
circ_0070203	miR-196b	GAS7	platinum sensitivity	-	[122]
circ_0007841	miR-532	NFIB	cisplatin resistance	-	[123]
circ_0026123	miR-543	RAB1A	cisplatin resistance	-	[124]
circ_0063804	miR-1276	CLU	cisplatin resistance	-	[125]
circ_0067934	miR-545	PPA1	-	cisplatin resistance	[126]
circ_Foxp1	miR-22	CEBPG	cisplatin resistance	-	[127]
	miR-150	FMNL3	cisplatin resistance	-	[127]
circ_NFIX	miR-942	NFIB	cisplatin resistance	-	[128]
circ_PLPP4	miR-136	PIK3R1	cisplatin resistance	-	[129]
circ_0000714	miR-370	RAB17, CDK6/RB	paclitaxel resistance	-	[130]
circ_0025033	miR-532	FOXM1	paclitaxel resistance	-	[131]
circ_0061140	miR-136	CBX2	-	paclitaxel sensitivity	[132]
circ_ATL2	miR-506	NFIB	paclitaxel resistance	-	[133]
circ_CELSR1	miR-149	SIK2	paclitaxel resistance	-	[134]
	miR-1252	FOXR2	paclitaxel resistance	-	[135]
circ_EXOC6B	miR-376c	FOXO3	-	paclitaxel sensitivity	[136]
circ_SETDB1	miR-508	ABCC1	paclitaxel resistance	-	[137]
circ_0000735	miR-526b	p-GP	docetaxel resistance	-	[138]
circ_0006404	miR-346	p-GP	-	docetaxel	[138]
circ_FURIN	miR-423	MTM1	testosterone effect	-	[139]
circ_MUC16	miR-199a	ATG13	autophagy	-	[140]
circ_RAB11FIP1	miR-129	DSC1	autophagy	-	[141]
circ_zinc finger	miR-212	superoxide dismutase 2	glycolysis	-	[142]

Via sponging of miRNAs by circRNAs, the cancer features continued to be either increased * or decreased **. Abbreviations are explained at the end of the article.

**Table 2 ncrna-10-00051-t002:** Interactions of circRNAs and miRNAs and their targets arranged to miRNAs which are sponged by two or more circRNAs.

circRNAs	miRNAs	Targets	Function		Ref.
			* Increasing	** Decreasing	
circ_CDK17	miR-22	CD147	proliferation, migration	-	[93]
circ_Foxp1		CEBPG	cisplatin resistance	-	[127]
circ_0007444	miR-23a	Dicer1	-	progression	[49]
circ_RHOBTB3		-	-	proliferation, invasion	[99]
circ_0005585	miR-23a/b	ESRP1	colonization	-	[112]
circ_0013561	miR-23b	ANXA2	EMT	-	[114]
circ_RAB11FIP1	miR-129	DSC1	autophagy	-	[141]
circ_SETDB1		MAP3K3	progression	-	[81]
circ_0000471	miR-135b	dusp5	-	progression	[46]
circ_CDR1		-	-	progression	[65]
circ_0061140	miR-136	CBX2	-	paclitaxel sensitivity	[132]
circ_PLPP4		PIK3R1	cisplatin resistance	-	[129]
circ_0009910	miR-145	-	progression	-	[52]
circ_0015756		PSAT1	progression	-	[53]
circ_ITCH		RASA1	-	progression	[71]
circ_VPS13C		MEK/ERK	cell cycle, motility	-	[101]
circ_WHSC1		MUC1	progression	-	[82]
circ_CELSR1	miR-149	SIK2	paclitaxel resistance	-	[134]
circ_PVT1		-	proliferation	-	[90]
circ_ATRNL1	miR-152	-	progression	-	[61]
circ_ZNF608		-	progression	-	[61]
circ_0004390	miR-198	-	proliferation	-	[88]
circ_MFN2		CUL4B	progression, glycolysis	-	[72]
circ_BNC2	miR-223	LARP4	-	progression	[62]
circ_BNC2		FBXW7	-	progression	[63]
circ_0025033	miR-370	SLC1A5	development	-	[119]
circ_0070203		TGFβR2	EMT	-	[113]
circ_0000714		RAB17, CDK6/RB	paclitaxel resistance	-	[130]
circ_ATRNL1	miR-378	Smad4	-	angiogenesis, metastasis	[108]
circ_IFNGR2		ST5	metastasis	-	[110]
circ_PLOD2		-	propagation	-	[102]
circ_0078607	miR-518a	Fas	-	progression	[60]
circ_NFIX		TRIM44	angiogenesis	-	[105]
circ_0007841	miR-532	NFIB	cisplatin	-	[123]
circ_0025033		FOXM1	paclitaxel resistance	-	[131]
circ_ATP2B4		SREBF1	metastasis	-	[109]
circ_0010467	miR-637	LIF/STAT3	platinum resistance	-	[121]
circ_0013958		PLXNB2	proliferation, invasion	-	[96]
circ_0051240		KLK4	migration, invasion	-	[103]
circ_EPSTI1	miR-942	-	progression	-	[67]
circ_PIP5K1A		NFIB	cisplatin resistance	-	[128]
circ_MUC16	miR-1182	S100B	progression	-	[73]
circ_WHSC1		hTERT	progression	-	[82]

By sponging of miRNAs by circRNAs, the cancer features continued to be either increased * or inhibited **. Abbreviations are explained at the end of the article.

**Table 3 ncrna-10-00051-t003:** CircRNAs as biomarkers and their relevance in prognosis.

circRNAs	Levels	Function		Associations	OC	Ref.
		Stimulation	Inhibition		Subtype	
circ_RS-7	up	-	-	FIGO stage	EOC	[158]
				lymph node		
				distant metastasis		
circ_HIPK3	up	proliferation, migration	apoptosis	FIGO stage	EOC	[159]
		invasion		lymph node		
circ_EXOC6B	up	-	-	FIGO stage	EOC	[160]
circ_N4BP2L2	up	-	-	FIGO stage	EOC	[161]
circ_RNA1656	down	-	-	FIGO stage	HGSOC	[162]
circ_0003972	down	-	-	-	EOC	[163]
circ_0007288	down	-	-	lymph node	EOC	[163]
circ_0078607	down	apoptosis	proliferation	-	HGSOC	[164]
circLARP4	down	-	-	FIGO, lymph node	EOC	[165]
circ-001567		proliferation	apoptosis	E-/N-cadherin	EOC	[166]
circ-NOLC1	up	proliferation, migration	-	FIGO stage	EOC	[167]
		invasion		differentiation		
circ_0013958	up	proliferation, migration	-	FIGO stage	EOC	[168]
		invasion		lymph node		
circ_BNC2	down	-	-	FIGO stage	EOC	[169]
				lymph node		
circ_SETDB1	up	relapse	-	FIGO stage	HGSOC	[170]
				lymph node		
circ-ABCB10	up	proliferation	apoptosis	FIGO stage	EOC	[171]
				differentiation		
				tumor size		

Up, upregulated; down, downregulated; HGSOC, high-grade ovarian serous carcinoma.

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
