# Peer review of "Interplay of microRNAs and circRNAs in Epithelial Ovarian Cancer"

_ncrna, 2024, doi:10.3390/ncrna10050051_

Round 1

Reviewer 1 Report

Comments and Suggestions for Authors

Heidi Schwarzenbach presents the review entitled “Interplay of microRNAs and circRNAs in Epithelial Ovarian 2 Cancer”. This is a well written and updated manuscript about the interplay of MiRNAs and circrnas in ovarian cancer. Although the review will be of interest for readers and deserves publication, I think that it could be improved if some changes mainly in the organization of data are added. Before potential publication of manuscript author should fully addresses several concerns.

Main concerns:

  1. For sections 3-5 It’s recommended to include descriptive two new figures summarizing several important sections of meta analysis. For instance a figure summarize the biogenesis and interplay between the MiRNAs and circnas including the novel mechanistic data reported for these molecules. 
  2. A second didactic figure must summarize the roles of described circrnas/mirnas pairs in ovarian cancer including the signaling and cellular processes they impact.
  3. For sections 5 and 6; it will be more accurate and informative to organize the role of the described MiRNAs and circrnas according to the cancer hallmarks they regulates (cell proliferation, apoptosis, migration, metastasis, etc).
  4. In table 1, the headings “enhancement” and “suppression” must be correctly aligned in the respective columns.

Author Response

Point by point reply to the comments raised by the reviewers with respect to the above manuscript. The reviewers’ comments are numbered and in regular font, while my reply is in italic and red. Modifications made in the revised manuscript were written in italic. I would like to thank the reviewer for the time and effort to provide the review. I have carefully and substantially revised the manuscript according to the suggestions provided.

 Main concerns:

  1. For sections 3-5 It’s recommended to include descriptive two new figures summarizing several important sections of meta analysis. For instance a figure summarize the biogenesis and interplay between the MiRNAs and circnas including the novel mechanistic data reported for these molecules.

I prepared, now a Figure 1 which shows an overview on the biogenesis of microRNAs and circRNAs, and their interaction influencing the translation. For a better overview, the important components was only shown (page 6).

  1. A second didactic figure must summarize the roles of described circrnas/mirnas pairs in ovarian cancer including the signaling and cellular processes they impact.

I prepared, now a Figure 2 which shows an overview on the circRNAs/miRNAs involved in the different signaling pathways (page 7).

  1. For sections 5 and 6; it will be more accurate and informative to organize the role of the described MiRNAs and circrnas according to the cancer hallmarks they regulates (cell proliferation, apoptosis, migration, metastasis, etc).

I arranged miRNAs and circrnas according to the cancer hallmarks (Table 1, pages 7-10 ).

  1. In table 1, the headings “enhancement” and “suppression” must be correctly aligned in the respective columns.

Sorry, I oversaw this shift. Now, it is on the right position (page 7).

Reviewer 2 Report

Comments and Suggestions for Authors

The manuscript titled "Interplay of microRNAs and circRNAs..." by Schwarzenbah provides a comprehensive and timely review of the processing, function, and application of miRNAs and circRNAs in epithelial cancer. The article thoroughly examines the interactions between circRNAs and miRNAs, as well as their experimentally validated molecular targets. Additionally, the author has meticulously tabulated the interactions of circRNAs and miRNAs, highlighting cases where two or more circRNAs sponge the same target. The manuscript also explores the potential of circRNAs as therapeutic agents and targets.

Although the biogenesis of miRNAs and circRNAs is briefly discussed, incorporating illustrations or figures to depict these processes would enhance readers' understanding. The article's extensive referencing further strengthens its impact, and the clarity of the English language used throughout makes it accessible to a broad audience within biomedical research. The well-organized tables present data clearly and concisely, adding significant value to the work. I commend the author for producing a high-quality and impactful review.

Author Response

Point by point reply to the comments raised by the reviewers with respect to the above manuscript. The reviewers’ comments are numbered and in regular font, while my reply is in italic. Modifications made in the revised manuscript were written in red. I would like to thank the reviewer for the time and effort to provide the review. I have carefully and substantially revised the manuscript according to the suggestions provided.

Although the biogenesis of miRNAs and circRNAs is briefly discussed, incorporating illustrations or figures to depict these processes would enhance readers' understanding. The article's extensive referencing further strengthens its impact, and the clarity of the English language used throughout makes it accessible to a broad audience within biomedical research. The well-organized tables present data clearly and concisely, adding significant value to the work. I commend the author for producing a high-quality and impactful review.

I prepared, now a Figure 1 which shows an overview on the biogenesis of microRNAs and circRNAs, and their interaction influencing the translation. For a better overview, the important components were only shown (page 6).

Reviewer 3 Report

Comments and Suggestions for Authors

The manuscript titled "Interplay of microRNAs and circRNAs in Epithelial Ovarian Cancer" by Heidi Schwarzenbach provides a comprehensive review of the literature on miRNAs and circRNAs, focusing on their interaction in the development, progression, and drug resistance of EOC. Additionally, the author highlights their potential as biomarkers and therapeutic targets. While the exposition is clear and the review is written well, I have some concerns, which are discussed below.

           1.     References are missing for some key findings, which need to be included throughout the manuscript. For example:

"Approximately 80% of the genome is transcribed to ncRNAs" (line 81)

"They play an essential role in various stages of cancer, namely initiation, progression, and metastasis" (line 85). Please provide references supporting the role of these ncRNAs in tumor initiation.

2.     In Table 1, the term "miRNA enhancement" is unclear. Please provide a definition and include it in the table legend. The table headings also need clarification. Additionally, ensure that the table data is specific to Epithelial Ovarian Cancer, as this aligns with the main theme of the article rather than covering all cancers.

3.     Line 122: Discuss some key miRNAs that act as oncogenes or tumor suppressors in Epithelial Ovarian Cancers. 

4.     Various researchers have identified numerous targets of these miRNAs across different pathways within the same tumor types. However, the manuscript does not adequately address the specificity of these miRNAs toward their intended targets, particularly considering the potential for miRNAs, like others in their class, to cause non-specific toxicity due to their broad range of targets. This lack of specificity could lead to off-target effects and unintended cellular responses. I recommend discussing the potential for non-specific toxicity and strategies to mitigate it, including how target specificity was ensured in this study. Adding this to the discussion section would provide essential context for interpreting the therapeutic potential and safety profile of these miRNAs.

5.     Please include a table summarizing the predictive biomarker potential of circRNAs in different types of Epithelial Ovarian Cancer. The table should also highlight their relevance in prognosis and correlation with treatment outcomes. This will provide a clear and concise overview of the clinical significance of circRNAs in Epithelial Ovarian Cancer, reinforcing the manuscript's key arguments regarding their role as potential diagnostic and therapeutic tools.

6.     I suggest that the authors include a figure illustrating the basic mechanism of miRNA regulation by circRNAs. This visual representation will help clarify the interplay between miRNAs and circRNAs and enhance the reader’s understanding of their regulatory roles.

 7. Are circRNAs better therapeutic targets than miRNAs in cancer?

Comments on the Quality of English Language

The quality is good.

Author Response

Point by point reply to the comments raised by the reviewers with respect to the above manuscript. The reviewers’ comments are numbered and in regular font, while my reply is in italic. Modifications made in the revised manuscript were written in red. I would like to thank the reviewer for the time and effort to provide the review. I have carefully and substantially revised the manuscript according to the suggestions provided.

  1. References are missing for some key findings, which need to be included throughout the manuscript. For example:

"Approximately 80% of the genome is transcribed to ncRNAs" (line 81)

I added ref. 15 (page 4, line 81).

"They play an essential role in various stages of cancer, namely initiation, progression, and metastasis" (line 85). Please provide references supporting the role of these ncRNAs in tumor initiation.

I added ref. 16 (page 4, line 85).

  1. In Table 1, the term "miRNA enhancement" is unclear. Please provide a definition and include it in the table legend. The table headings also need clarification. Additionally, ensure that the table data is specific to Epithelial Ovarian Cancer, as this aligns with the main theme of the article rather than covering all cancers.

In Table 1 and 2, I changed “enhancement” into “increasing”, and provided the definition below the tables. In added EOC to the headings (pages 7, 10, 12, 13).

  1. Line 122: Discuss some key miRNAs that act as oncogenes or tumor suppressors in Epithelial Ovarian Cancers.

Now, I mentioned some miRNAs and those function in EOC as examples (page 5).

  1. Various researchers have identified numerous targets of these miRNAs across different pathways within the same tumor types. However, the manuscript does not adequately address the specificity of these miRNAs toward their intended targets, particularly considering the potential for miRNAs, like others in their class, to cause non-specific toxicity due to their broad range of targets. This lack of specificity could lead to off-target effects and unintended cellular responses. I recommend discussing the potential for non-specific toxicity and strategies to mitigate it, including how target specificity was ensured in this study. Adding this to the discussion section would provide essential context for interpreting the therapeutic potential and safety profile of these miRNAs.

I discussed this aspect in the conclusion (page 18).

  1. Please include a table summarizing the predictive biomarker potential of circRNAs in different types of Epithelial Ovarian Cancer. The table should also highlight their relevance in prognosis and correlation with treatment outcomes. This will provide a clear and concise overview of the clinical significance of circRNAs in Epithelial Ovarian Cancer, reinforcing the manuscript's key arguments regarding their role as potential diagnostic and therapeutic tools.

I added Table 3 with a separate paragraph “8. CircRNAs as predictive biomarkers” to the manuscript (pages

  1. I suggest that the authors include a figure illustrating the basic mechanism of miRNA regulation by circRNAs. This visual representation will help clarify the interplay between miRNAs and circRNAs and enhance the reader’s understanding of their regulatory roles.

I added Figure 1 to the manuscript (page 15, 16).

  1. Are circRNAs better therapeutic targets than miRNAs in cancer?

It is difficult to say whether circRNAs or miRNAs are better therapeutic targets. Investigative studies can only answer this question. At first glance, circRNAs appear to be the better therapeutic targets because they are above miRNAs and regulate their function. However, when circRNAs as therapeutics are targeted, their binding to miRNAs is abrogated, resulting in off-target effects because of the binding of the free miRNA to a broad range of targets. This may lead to potential non-specific toxicity. Therefore, it appears that miRNAs are better therapeutics, because the inhibition of miRNAs only comprises one molecule type that of course can influence numerous downstream components, whereas the inhibition of circRNAs comprises two molecule types: the inhibition of circRNAs accompanied by the abrogation of the binding to their miRNA targets.

Round 2

Reviewer 1 Report

Comments and Suggestions for Authors

The authors have answered all my concerns; thus, I recommend accepting the manuscript for publication in this actual form.